# Effect of Alternating Current and Sulfate-Reducing Bacteria on Corrosion of X80 Pipeline Steel in Soil-Extract Solution

**DOI:** 10.3390/ma12010144

**Published:** 2019-01-04

**Authors:** Yongchang Qing, Yunlong Bai, Jin Xu, Tangqing Wu, Maocheng Yan, Cheng Sun

**Affiliations:** 1Environmental Corrosion Center, Institute of Metals Research, Chinese Academy of Sciences, Shenyang 110016, China; ycqing15s@imr.alum.ac.cn (Y.Q.); ylbai@163.com (Y.B.); yanmaocheng@imr.ac.cn (M.Y.); 2Institute of Petrochemical Technology, Shenyang University of Technology, Liaoyang 111003, China; 3School of Materials Science and Engineering, Xiangtan University, Xiangtan 411105, China; tqwu10s@imr.ac.cn

**Keywords:** alternating current, sulfate-reducing bacteria, X80 pipeline steel, soil-extract solution

## Abstract

AC corrosion has been considere d as a threat to the corrosion of buried pipelines. Effects of sulfate-reducing bacteria (SRB) and alternating current (AC) on corrosion of X80 pipeline steel in soil-extract solution were investigated by electrochemical and surface analysis techniques. AC current can inhibit the growth of planktonic and sessile SRB. The corrosion current density of steel with 10 mA/cm^2^ AC current is about nine times bigger than that without AC current. Corrosion morphology changes from small pitting to large pitting holes with increasing AC current density. Corrosion of steel with SRB and AC current is controlled by both active dissolution of iron and film degradation.

## 1. Introduction

Stray currents caused by high voltage facilities, such as high power transmission lines and electrified railways, can enhance the corrosion of buried pipeline steels, eventually resulting in significant failures of the pipelines [1,2,3]. Studies of alternating current (AC) influenced corrosion can be traced back to the early period of the 1900s. These early investigations showed that the corrosion rate of metal induced by AC current was approximately 1% of that by an equivalent amount of direct current (DC), [4] However, AC influenced corrosion had not been considered as a threat to the integrity of pipelines until a pipeline accident caused by AC current occurred in Germany in 1986, stimulating interest in this research topic [3,5].

Large numbers of studies have been performed to analyze effects of AC currents on the corrosion of metals [6,7,8,9]. Goidanich et al. found that AC current density higher than 10 A/m^2^ gave rise to a considerable increase in the corrosion rate of the metal. When the AC current density was higher than 500 A/m^2^, the corrosion rate was greater than 1 mm/y [10,11]. The results by Guo et al. showed that the corrosion rate of X series pipeline steel increased and the corrosion potential shifted negatively with increasing AC current density [6,8]. Some studies also indicated that the AC current caused the corrosion potential of the pipeline steel to shift negatively [3,7,12]. It was reported that the AC current could change the corrosion morphologies of the metals [6,7,8,10,11,12,13]. Guo and Fu et al. [8,12] found that uniform corrosion occurred on X60 steel surface when the AC current density was small, while pitting holes were formed at high AC current density. They concluded that the AC current enhanced localized corrosion of the steels. Li et al. also reported that the corrosion morphologies of the metals changed from uniform corrosion to pitting corrosion once the AC current was applied, and further transformed to etching moats with increasing AC current densities [13]. They also found that alternating voltage had a strong influence on the corrosion of Q235 steels, and the corrosion rate increased rapidly with increasing voltage. Guo et al. also investigated the effect of AC frequency on the corrosion of the pipeline steel in simulated soil solutions, and concluded that the Faradaic current caused by the AC current decreased with increasing AC frequency, which inhibited the corrosion of the steel [6,8].

Other factors have also been considered in studying the effect of the AC current on the corrosion of the metals. Tang et al. investigated the effects of pH values on the AC corrosion of carbon steels. They found that the corrosion rate of the steel decreased with increasing pH value. It was due to the fact that a protective passive film was formed on the steel surface at a high-pH condition, however, the film suffered from mechanical breakdown once the AC current was applied to the steel, and the protective ability of the film to the steel was lost gradually leading to an increase of corrosion rates [1]. Kuang and Guo et al. studied the effect of cathodic protection potential on the AC corrosion of pipeline steel, and concluded that the AC current decreased cathodic protection effectiveness of the steels, however, the cathodic protection potential which was more negative than −950 mV _vs. CSE_ (copper sulfate electrode) could relieve corrosion risks induced by the AC current [14,15].

Microbiologically influenced corrosion of metals is widely found in different environments [16,17,18,19,20], however, the effects of AC current on microbiologically influenced corrosion are seldom studied. Sulfate-reducing bacteria (SRB) are the most important microbes for the corrosion of buried pipelines, and SRB influenced corrosion has been widely found in soils, mines, seaports, lakes, power plants, sewage treatment facilities, offshore structures, and so on [21,22]. In this study, the effects of the AC current on SRB influenced corrosion of X80 pipeline steel were investigated in soil-extract solutions (SES).

## 2. Materials and Methods

### 2.1. Coupon Preparation

X80 pipeline steel was used, complying with standard API SPEC 5LX80, which has a nominal composition (w/w%) of C 0.070, Mn 1.820, P0.007, S 0.023, Si 0.190, Mo 0.010, Ni 0.170, Cr 0.026, Cu 0.020, V 0.002, Nb 0.056, Ti 0.012, Al 0.028, and Fe balance. It was cut into square coupons (10 mm × 10 mm × 3 mm) used as working electrodes. Coupon preparation was controlled carefully to ensure that there was no bubble and groove generated at the epoxy/steel interfaces by using a 100 times magnifying glass. The prepared coupons were sequentially ground with a series of SiC papers (150, 400, 600, 800, 1000, and 2000 grit). They were then washed sequentially with deionized water and 100% ethanol (Sinopharm Chemical Reagent limited corporation, Shenyang, China) to clean and degrease the coupon surfaces. The coupons were dried and kept in a desiccator (Sinopharm Chemical Reagent limited corporation, Shenyang, China). The working electrodes were sterilized under ultraviolet light for 30 min prior to experiments.

### 2.2. Soil-Extract Solution Preparation

The soils used in this study were taken from Shenyang, China. The SES was prepared by extracting soil solution with a water-soil ratio of 5:1. Analytical results of solution compositions are given in Table 1. The SES was autoclaved at 121 °C for 20 min and stored at 4 °C in a refrigerator for use. The SES was deoxygenated using high-pure N_2_ sparging for at least 2 h before experiments, and the dissolved oxygen concentration was less than 1 ppm.

### 2.3. Microorganisms

Sulfate reducing bacteria (SRB) strains, *Desulfovibrio desulfuricans*, were used as described in a previous paper [23], and anaerobically incubated in the API RP-38 medium (g/L), containing MgSO_4_·7H_2_O (Sinopharm Chemical Reagent limited corporation, Shenyang, China) 0.2; ascorbic acid (Sinopharm Chemical Reagent limited corporation, Shenyang, China) 1.0, NaCl (Sinopharm Chemical Reagent limited corporation, Shenyang, China) 10.0, KH_2_PO_4_ (Sinopharm Chemical Reagent limited corporation, Shenyang, China) 0.5, sodium lactate (Sinopharm Chemical Reagent limited corporation, Shenyang, China) 4.0, yeast extract 1.0, Fe(NH_4_)_2_(SO_4_)_2_ (Sinopharm Chemical Reagent limited corporation, Shenyang, China) 0.02.

Fifty mL of SRB culture medium was subsequently transferred into 950 mL of the sterilized SES. After 2 days incubation, the inoculated SES was added in the testing devices for the experiments

The most probable number (MPN) method, according to the American Society of Testing Materials (ASTM) Standard D 4412-84, was used to count SRB planktonic cells. A series of 1:10 serial dilutions was carried out using an SRB culture medium. The small vials in MPN were then incubated at 30 °C. The vials with viable SRB cells showed black FeS precipitation. For sessile cell enumeration, the biofilm on a coupon surface was first scraped off into a 7.4 PBS solution and then vortexed for 30 s to get a cell suspension for MPN. The MPN cell count in the solution was converted to cells·cm^−2^ based on the coupon surface area of 1 cm^2^.

### 2.4. Tafel Curve

Electrochemical measurements were carried out using a PARSTAT2273 electrochemical workstation (AMETEK Instruments, USA) in a three electrode cell system, where the steel coupon was used as a working electrode (WE), a platinum plate as a counter electrode (CE), and a saturated calomel electrode (SCE) as a reference electrode (RE), as shown in Figure 1a. A function generator of UTG-9000A model was used to provide an AC signal to the coupons. A sine AC waveform with a frequency of 50 Hz was applied between the working electrode and a graphite electrode. A rheostat was used to adjust the AC current density. An inductor (5 H) was introduced to avoid interference of the AC signal applied to the electrochemical workstation, while a capacitor (25 V, 470 μF) was used to prevent DC flowing into AC power. A peristaltic pump was used to renew SES, providing nutrients for the growth and metabolism of SRB in a long-term experiment. High purity nitrogen was used to remove oxygen in SES during the whole experiment.

Four conditions including the sterilized SES solution without AC current (Control), the inoculated SRB solution without AC current (SRB), the inoculated SRB solution with AC current density of 5 mA/cm^2^ (SRB + 5 mA/cm^2^ AC), and the inoculated SRB solution with AC current density of 10 mA/cm^2^ (SRB + 10 mA/cm^2^ AC) were tested.

Tafel curves were measured at a potential sweeping rate of 1.0 mV/s under the various AC current densities, and the scanning range was from −250 mV to +250 mV (vs. the open circuit potential). All tests were operated at 25 ± 1 °C.

### 2.5. Weight Loss

Each cell in the weight loss testing described below contained 3 replicate coupons for weight loss measurements, and each coupon with a total exposed surface area of 20 cm^2^ was hung in the SES solutions, vertically connecting with a thin copper wire by a screw, as shown in Figure 1b. No-working areas were sealed by lacquer. The coupons were retrieved after 30 days of experiment, and the lacquer on the surface of the metals was removed by paint stripper. A descaling solution (HCl, ρ = 1.19 g·mL^−1^, 100 mL; hexamethylenetetramine, 0.35 g; distilled water, 100 mL) was used to remove biofilms and corrosion products from coupon surfaces. A fresh descaling solution was used every time. The acid attack effect during cleaning was not significant compared to weight losses by microbiologically influenced corrosion (MIC). After cleaning, the coupons were rinsed with 100% ethanol and dried before weighing. Three coupons were weighed to obtain the average for each weight loss data point.

### 2.6. Morphological Characterization

The morphologies of corrosion products and the pitting morphologies of the coupon surfaces were characterized using scanning electron microscopy (SEM, XL30-FEG) (Philips, Holand) and atom force microscopy (AFM, Pico Plus AFM) (Digital Instruments, USA) images. The coupons with corrosion products were retrieved from the cells in an anaerobic chamber and washed three times with phosphate-buffered saline (PBS) solutions to remove planktonic cells and culture medium residues. Then, the coupons were immersed for 4 h in a 3% glutaraldehyde solution to fix biofilms followed by dehydration using 25%, 50%, 75%, 95%, and 100% ethanol solutions sequentially (10 min each). Finally, the biofilms were dried using a supercritical CO_2_ dryer followed by a palladium coating to provide conductivity before SEM analysis.

The corrosion products formed on the coupon surfaces were removed carefully by both mechanical and chemical methods. The mechanical methods, including light scraping and scrubbing, were used for removal of tightly adherent corrosion products. In the chemical procedure, the descaling solution was used, and then the steel coupons were washed with deionized water and 100% ethanol before SEM observation.

## 3. Results

### 3.1. SRB Cell Counts

Figure 2 shows variations of planktonic SRB cell counts in SES after 2 days, 10 days, 20 days, and 30 days and sessile SRB cell counts after 30 days under three conditions (SRB, SRB + 5 mA/cm^2^ AC and SRB + 10 mA/cm^2^ AC). For the solutions with only SRB and SRB + 5 mA/cm^2^ AC, the MPN planktonic SRB cell counts are the same in the whole period of the experiment. The cell counts decreased from 9.5 × 10^7^ cells/mL on the second day to 4.5 × 10^7^ cells/mL on the 10th day, and then remained stable after 10 days. For the solution with SRB + 10 mA/cm^2^ AC, the planktonic cell counts were only 2.5 × 10^7^ cells/mL on the second day. The cell counts decreased with time, and reach 9.5 × 10^6^ cells/mL on the 10th day. Finally, the cell counts were reduced to 4.5 × 10^6^ cells/mL after 30 days. The above results show that the AC current density of 5 mA/cm^2^ has little effect on the planktonic SRB cells in the SES solution, however, the AC current density of 10 mA/cm^2^ can obviously inhibit the growth and metabolism of the planktonic SRB cells.

It can also be seen from Figure 2 that the MPN sessile cell counts are 2.5 × 10^8^ cells·cm^−2^ on the surface of the X80 pipeline steel coupon in the presence of only SRB after 30 days. The AC current density of 5 mA/cm^2^ achieved about 2-log reduction in SRB sessile cell count, reducing to 9.5 × 10^6^ cells/cm^2^, indicating that the 5 mA/cm^2^ AC current has an ability to prevent sessile SRB cells after 30 days of exposure. When the AC current density is up to 10 mA/cm^2^, the sessile SRB cell count is further reduced to 9.5 × 10^4^ cells/cm^2^, achieving an extra 2-log reduction compared to 5 mA/cm^2^ AC current treatment. The results showed that the AC current also inhibits the sessile SRB cell growth, and inhibition increases with increasing AC current density.

### 3.2. Micro-Analysis of the Coupon Surface

SEM corrosion product images of X80 pipeline steel coupon surface after 30 days are shown in Figure 3. For the control condition, the steel coupon surface was partially covered by a thin layer of corrosion products after 30 days, as shown in Figure 3a. A layer of dense and thick composite film including corrosion products and biofilms was observed on the surface of the steel coupon, as seen in Figure 3b. When the AC current density of 5 mA/cm^2^ was applied, some crater-shaped corrosion morphologies were observed on the coupon surface (Figure 3c), which may be due to generation of gas bubbles under the effects of AC current [9,10,11]. This leads to a decrease of the protective ability of the film. With the AC current density increasing to 10 mA/cm^2^, the film becomes loose, as shown in Figure 3d, and some flocculent corrosion products can be seen on the steel coupon surface. The result indicates that the AC current has an influence on the film formed on the steel coupon surface. The film becomes looser and looser on increasing the AC current density.

It can also be seen from Figure 3 that there are a large number of SRB sessile cells, as pointed out by red arrows, on the surface of the X80 pipeline steel coupon without the AC current in the presence of SRB (Figure 3b). The SRB sessile cells observed on the coupon surface with AC current density of 5 mA/cm^2^ are still abundant, as shown in Figure 3c. When the AC current density is up to 10 mA/cm^2^, the SRB sessile cells are much less than those in the SRB + 5 mA/cm^2^ AC condition (Figure 3d). This indicates that the AC current (e.g., 10 mA/cm^2^) obviously inhibits the growth and metabolism of SRB sessile cells on the surface of the steel coupon. It is in accordance with the results of the MPN sessile cell counts.

Table 2 shows EDAX results of the corrosion products of X80 steel coupons after 30 days in the four conditions (control, SRB, SRB + 5 mA/cm^2^ AC, and SRB + 10 mA/cm^2^ AC). The corrosion products contain the elements C, O, Ca, Si, and Fe in four conditions. Elements O and Fe come from the corrosion products, and elements Ca and Si from soil particles. There are also elements P and S in the presence of SRB, which are mainly from the metabolites of SRB. Element S slowly decreases from 5.8% (At%) to 5.58% when the AC current density of 5 mA/cm^2^ is applied, and sharply decreases to 3.39% with the current density increasing to 10 mA/cm^2^. Element P also has the same changing tendency as element S. It indicates that the AC current has an influence on the metabolism of SRB.

SEM cross-sectional images of the X80 pipeline steel coupons in the four conditions are shown in Figure 4. The corrosion products are partially distributed on the surface of the steel coupon for the control coupon after 30 days, as seen in Figure 4a. In the presence of SRB, a thick and dense composite film is observed on the surface of the steel coupon (Figure 4b). When an AC current density of 5 mA/cm^2^ is applied, the film becomes loose, as shown in Figure 4c, and there are some holes in the film. Only a few corrosion products can be seen with the AC current density up to 10 mA/cm^2^, and some deep pitting holes are formed (Figure 4d).

Figure 5 shows the corrosion morphologies of X80 pipeline steel coupons after removing the corrosion products and biofilms after 30 days in the four conditions (control, SRB, SRB + 5 mA/cm^2^ AC and SRB + 10 mA/cm^2^ AC). For the control coupon, the steel coupon is slightly corroded because of corrosivity of the SES solution after 30 days (Figure 5a), and no pitting hole is observed on the surface of the steel coupon. In the presence of only SRB, some small pitting holes are observed on the steel coupon surface after 30 days, as shown in Figure 5b. With the application of the 5 mA/cm^2^ AC current, numbers of the pitting holes formed on the coupon surface increase compared with the condition without the AC current, and the sizes of the holes were much larger than those without AC current (Figure 5c). When the AC current density is up to 10 mA/cm^2^, the corrosion of the steel coupon becomes more severe, as shown in Figure 5d, and the corrosion morphology changes from small pitting holes to large pitting holes. It is different from the AC influenced corrosion in the absence of SRB reported by other researchers [8,23]. They found that the low AC current density led to uniform corrosion, while the high AC current density caused pitting corrosion. They concluded that AC oscillation between a steel coupon and an electrolyte solution caused the pitting corrosion of the steel electrode. Wakelin’s result [24] showed that the corrosion did not occur on the steel surface at the low AC current density, while severe corrosion could be observed if the AC current density was greater than 10 mA/cm^2^.

For further investigation of the effects of AC current on the corrosion of X80 pipeline steel coupons, an AFM analysis was used. Figure 6 and Figure 7 show AFM images of X80 pipeline steel coupons after removing the corrosion products and biofilms after 30 days in the four conditions (Control, SRB, SRB + 5 mA/cm^2^ AC and SRB + 10 mA/cm^2^ AC), and the analysis data are given in Table 3. The steel coupon surfaces become rougher and rougher with increasing AC current density (Figure 6). As shown in Table 3, the mean roughness of the steel coupon surface increases from 43 nm to 78 nm with addition of the SRB cells into the SES solution. When the AC current density of 5 mA/cm^2^ is applied, the mean roughness of the surface further increases to 136 nm. With the AC current density higher up to 10 mA/cm^2^, the mean roughness sharply increases to 278 nm, reaching nearly seven times of that without the AC current and SRB. It can also be seen from Table 3 that R_p-v_ (a different value between peak and valley) value is only 582 nm for the control coupon. The R_p-v_ value increases to 1177 nm in the presence of SRB. The R_p-v_ value is up to 1676 nm with the application of the AC current density of 5 mA/cm^2^, and further increases to 1984 nm when the AC current density increases to 10 mA/cm^2^. The above results prove that AC current can enhance the localized corrosion of steel coupon, increasing the depth of the pitting holes.

### 3.3. Tafel Curves

Figure 8 shows polarization curves of X80 pipeline steel coupons measured under different conditions after different days, and the fitting results are given in Table 4. Anodic current densities increase and corrosion potentials are shifted negatively with the AC current density changing from 0 to 10 mA/cm^2^. However, cathodic current densities decrease with the application of AC current. The increase of anodic current density indicates that anodic dissolution of X80 steel coupon is enhanced when the AC current is applied, and the anodic oxidation rate of the steel coupon increases with increasing AC current density. The decrease of the cathodic current density shows that cathodic reaction is inhibited by AC current. It can been seen from Table 4 that the corrosion current density sharply increases when an AC current density of 5 mA/cm^2^ is applied, especially after 10 days, and further increases on the current density increasing to 10 mA/cm^2^. The corrosion current density of the steel coupon decreases with time in the absence of AC current, but increases with time under the effect of AC current. The corrosion current density of the steel coupon with the AC current density of 10 mA/cm^2^ is about nine times bigger than that without the AC current after 30 days. The results show that the application of the AC current can enhance the SRB influenced corrosion of X80 pipeline steel coupon.

### 3.4. Weight Loss

Corrosion rate calculated by weight loss was used to validate the electrochemical results, as shown in Figure 9. The corrosion rate is slightly higher in the presence of SRB than that without SRB, increasing from 0.0314 ± 0.0058 mm/y to 0.0329 ± 0.0072 mm/y. With the application of the 5 mA/cm^2^ AC current, the corrosion rate of the steel coupon sharply increases to 0.2105 ± 0.0116 mm/y. The corrosion rate further increases to 0.3059 ± 0.0096 mm/y when the AC current density is 10 mA/cm^2^, almost three times of that of the control coupon. It indicates that the AC current can severely enhance the corrosion of the X80 steel coupon, which is in accordance with the result of the electrochemical measurement.

## 4. Discussion

The anodic dissolution reaction of metal is mainly determined by the anodic current density. When the anodic and cathodic reactions of metal reach an equilibrium, the value of the anodic current density of the metal is equal to that of the cathodic current density. Once the AC current is applied to this stable system, the balance between the anodic and cathodic reactions is destroyed. Because of nonlinearity of the AC current, the increase of the anodic current is larger in the positive half cycles of the AC current than that of the cathodic current in the negative half cycles. In the whole AC cycle, a net increase of anodic current is obtained. Additionally, from the view of thermodynamics, the metal oxidation easily proceeds during the positive half cycles of AC current compared to reduction of metal oxide in the negative half cycles, which leads to the acceleration of metal corrosion [10,11].

At ambient temperatures in neutral or near-neutral solution containing dissolved oxygen, the corrosion of iron is driven by the dissolved oxygen concentration. Anodic reaction takes place as follows:(1)$$\mathrm{Fe}\to {\mathrm{Fe}}^{2+}+2e^{-}$$

Oxygen is reduced at cathodic regions:(2)$$O_{2}+4e^{-}+2H_{2}O\to 4{\mathrm{OH}}^{-}$$

For the SRB influenced corrosion, the oxygen reduction reaction does not occur because of anaerobic conditions. Van Wolzogen Kühr mentioned that the cathodic process of iron corrosion in the presence of SRB in anaerobic soils is represented by the following equation [25]:(3)$${\mathrm{SO}}_{4}^{2-}+8H^{+}+8e^{-}\overset{\mathrm{SRB}}{\to }S^{2-}+4H_{2}O$$

In the anaerobic condition, the metal is thought not to be corroded without oxidizing agents because the cathodic reaction cannot occur. When SRB is added, Equation (3) can take place in the presence of sulfate ions, and this cathodic reaction is not related to the dissolved oxygen concentration. As a result, the corrosion of metal can occur in the presence of SRB even though there are not any oxidizing agents. With the application of AC current, the anodic and cathodic reactions of metal corrosion must be influenced. At the same time, the AC current also affects the growth of SRB, as shown in Figure 2 and Figure 3, which indirectly influences the corrosion of metal.

As shown in Figure 8, the cathodic current density slowly decreases with the AC current density increasing from 0 to 5 mA/cm^2^, and sharply decreases when the current density is up to 10 mA/cm^2^, especially in the later period of the whole experiment. The result indicates that the application of AC current inhibits the cathodic reaction. It can be seen from Equation (3) that the cathodic reaction is controlled by the growth and metabolism of SRB. The growth and metabolism of SRB becomes slower and slower under the effect of AC current, as seen in Figure 2, which inhibits the cathodic reaction of the steel corrosion.

The anodic current density of steel coupon increases with the increase of AC current density. It indicates that the AC current accelerates the dissolution rate of steel, leading to the increase of the anodic current density of the steel coupon corrosion. A rectification mechanism was developed by McCollum [26]. He pointed out that a net Faraday current was rectified for asymmetry of anodic and cathodic polarization when the metal experienced the anodic and cathodic polarization under the effect of AC current. Goidanich et al. attributed the AC influenced corrosion to the irreversibility of the anodic reaction [11]. It is considered that the metal ions generated from the oxidization process of metal in the positive half cycles of AC current cannot be reduced to metal atoms completely in the negative half cycles. From the thermodynamic view, the metal dissolution reaction is the most favored process during the anodic half cycles, while the reactions of oxygen reduction and hydrogen evolution are favored during the cathodic half cycles. The reasons mentioned above lead to the increase of the anodic current density of the steel corrosion under the effect of AC current.

It can also been seen from Table 4 that, for the condition without AC current, the corrosion current density of the steel coupon decreases with time. This is due to the fact that a layer of composite film containing the corrosion products, biofilm and metabolic sulfide is formed on the surface of the steel coupon. This film is dense, compact, and protective, as shown in Figure 3. This result is consistent with Sheng and González’s findings [27,28]. It is well known that a film containing FeS sometimes offers protection against corrosion [29,30]. Castaneda and El Mendili et al. found iron sulfide could enter into the pores of the corrosion product film, and with gradual accumulation of the iron sulfide the film became denser and denser, which decreased the corrosion rate [31,32].

When the AC current is applied to the steel coupons, some of the current is involved in electrode reaction, acting as Faradaic current, and the rest takes part in charging–discharging processes, acting as non-Faradaic current, in an electric double layer between the electrode surface and the electrolyte solution. It is known that when the electrode is immersed in an electrolyte solution, the balance of the electrode reaction is achieved in the absence of AC interference. Once AC interference is applied, the Faradaic current caused by the AC current will polarize the steel electrode, and the original balance of the electrode reactions is destroyed by this polarization, and then a new balance is reached.

There are two reasons causing enhancement of steel corrosion, Fe active dissolution and film degradation.

(1) Fe active dissolution

When an iron electrode is immersed in an electrolyte solution, an electric double layer is formed on the iron surface. Electrons are on the side of the iron, and cationic ions on the side of the solution. When the AC current is applied to the iron, the Faradaic current accelerates the dissolution of Fe atoms in the positive half cycles of AC current, forming ferrous ions. These ferrous ions diffuse into the bulk solution under the effect of an electric field between the working electrode and the counter electrode. In order to reach the new balance, the dissolution of the iron further increases due to the decrease of concentration of the ferrous ions on the iron/solution interface. Kuang and Cheng also found that the electric field caused by the AC current could increase the diffusing rate of ferrous ions [16].

(2) Film degradation

The corrosion product layer generated by the active dissolution of iron can be a barrier to prevent ferrous ions from flowing into the bulk solution and corrosive anions from reaching the iron/corrosion product interface. When the AC current is applied, the corrosion product layer on the surface of the iron is changed, which further influences the corrosion of iron.

Researches have shown that the structure of films on a metal surface could be changed or destroyed under the effect of AC current [33]. Sato thought that breakdown of the film resulted from a sudden change of the metal potential caused by the presence of the electric fields across the film [34]. Goidanich et al. indicated that the AC current led to the growth of corrosion product film which was thick but non-adherent on the metal surface. The nature, thickness, and adhesion of the film changed with the AC current density [10,11]. Tang et al. also found that the application of AC current changed the state of the passive layer which was reduced to form a porous rust layer [1,2]. They also concluded that the passive film suffered from mechanical breakdown caused by the AC electric field across the film.

Other studies show that the corrosion product layer can be taken from the iron surface by the massive quantities of gases generated from the cathodic reactions in the negative half cycles of AC current. Ghanbari et al. found that with the polarization potential becoming more negative, hydrogen bubbles were formed on the coupon surface [35]. Xu et al. thought that the vast majority of the AC current either participated in the charging–discharging process (acting as non-Faradaic current) of the double-charge layer or became involved in the redox reaction of water [9]. They also found that the high AC current density could generate a high instantaneous potential of the steel coupon which was more negative than the hydrogen evolution potential, producing hydrogen atoms, and then generating large quantities of small bubbles on the electrode surface due to the combination of the hydrogen atoms. Crater-shaped morphologies were observed on the surface of the steel coupon, formed due to the generation of the gas bubbles, as shown in Figure 3c.

A model was developed to illustrate the effect of AC current on the corrosion of X80 pipeline steel, as shown in Figure 10. In the absence of AC current, a composite film containing the corrosion products and biofilms is formed on the steel surface. This composite film is protective. The protective ability of the film becomes stronger and stronger as metabolite sulfide particles, such as FeS, enter into tunnels, film defects, as illustrated in Figure 10a. This blocks diffusion of corrosive ions onto the steel/corrosion product interface through the film, and ferrous ions into the bulk solution. An electrical double layer (EDL) is formed on the steel/solution interface when the iron electrode is immersed in an electrolytic solution. The positive and negative charges are equal on either side of the EDL when the electrode system reaches an equilibrium state. There are quantities of electrons on the side of the steel, and cationic ions, such as Fe^2+^, on the side of the solution.

In the positive half cycles of AC current, the AC current is considered as an electrolytic current leading to the anodic dissolution of iron. At the same time, an electric field is formed between the working electrode and the counter electrode. The ferrous ions adsorbed on the interface diffuse into the bulk solution under the effect of the electric field, and the charge balance of the EDL is destroyed. More ferrous ions are inevitably generated by the Fe oxidation reaction in order to reach a new equilibrium of the EDL, as shown in Figure 10b. This increases the corrosion rate of steel coupon.

In the negative half cycles of AC current, the gas bubbles, mainly hydrogen gas, are generated on the interface of the electrode and film under the effects of the cathodic currents of AC current. Gas pressures induced by these bubbles in the film will destroy the film. The more profuse the gas bubbles are, the more is the breakage of the film, as shown in Figure 10c. Finally, the film is partially detached. Furthermore, the corrosion products, such as iron oxide, may also be reduced by the cathodic current of the AC current, resulting in a decrease of film thickness. This further weakens the protective ability of the film, indirectly enhancing the corrosion of the steel coupon.

## 5. Conclusions

The effect of AC current on the behavior and mechanism of SRB influenced corrosion of X80 pipeline steel coupon was investigated in soil-extract solutions by electrochemical measurement and surface analysis technology. The conclusions are as follow:(1)The 10 mA/cm^2^ AC current inhibits the growth and metabolism of planktonic and sessile SRB cells, indirectly enhancing the corrosion of the steel coupon.(2)Tafel results show that the AC current can enhance the corrosion of X80 pipeline steel coupon, and the corrosion current density of steel coupon with an AC current density of 10 mA/cm^2^ is almost 10 times as much as that without the AC current.(3)The gas bubbles generated in the negative half cycles of AC current result in the formation of crater-shaped corrosion morphologies. There is a significant difference in the corrosion behavior of the steel coupon with and without AC application in the presence of SRB. Even when the AC current density is only 5 mA/cm^2^, severe pitting corrosion still occurs. The corrosion type of steel coupon changes from pitting corrosion to localized corrosion with the increase of AC current density.(4)The mechanism of AC influenced corrosion in the presence of SRB is controlled by both iron active dissolution and film degradation.

## Figures and Tables

**Figure 1 materials-12-00144-f001:**
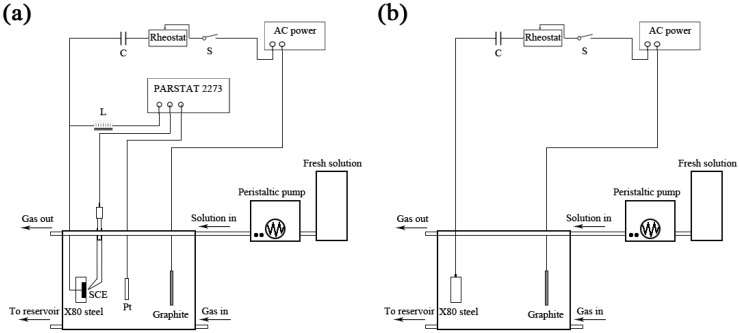
Schematic diagram of the electrochemical (**a**) and weight-loss (**b**) experimental setup of the effect of sulfate-reducing bacteria (SRB) and alternating current (AC) on the corrosion of X80 pipeline steel coupon in the soil-extract solution.

**Figure 2 materials-12-00144-f002:**
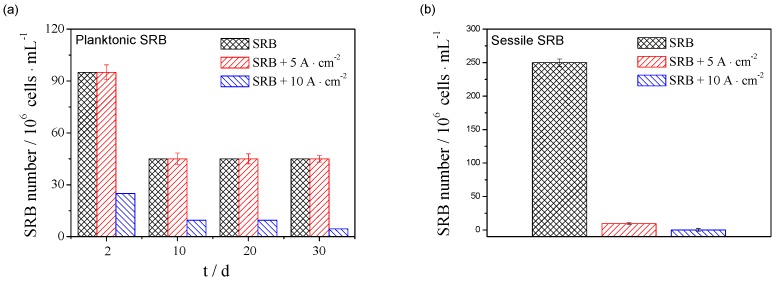
Variations of planktonic SRB numbers (**a**) with time, and sessile SRB numbers (**b**) after 30 days in the presence of SRB, SRB + 5mA/cm^2^ AC, and SRB + 10mA/cm^2^ AC.

**Figure 3 materials-12-00144-f003:**
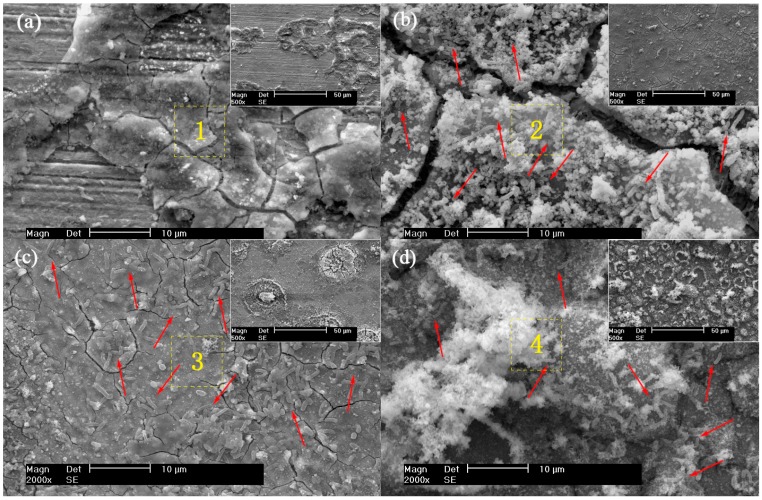
SEM corrosion product images of X80 pipeline steel coupons after 30 days. (**a**) Control; (**b**) SRB; (**c**) SRB + 5 mA/cm^2^ AC; (**d**) SRB + 10 mA/cm^2^ AC.

**Figure 4 materials-12-00144-f004:**
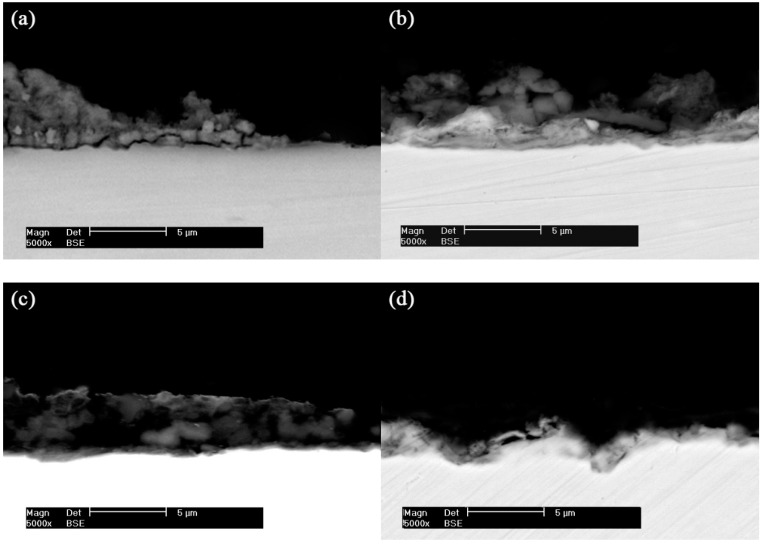
SEM cross-sectional images of X80 pipeline steel coupons after 30 days. (**a**) Control; (**b**) SRB; (**c**) SRB + 5 mA/cm^2^ AC; (**d**) SRB + 10 mA/cm^2^ AC.

**Figure 5 materials-12-00144-f005:**
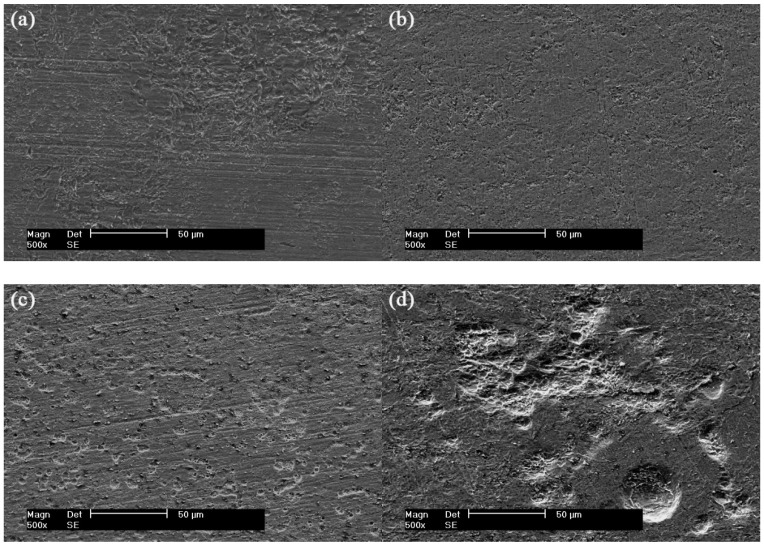
SEM images of X80 pipeline steel coupons after removing corrosion products and biofilms after 30 days. (**a**) Control; (**b**) SRB; (**c**) SRB + 5 mA/cm^2^ AC; (**d**) SRB + 10 mA/cm^2^ AC.

**Figure 6 materials-12-00144-f006:**
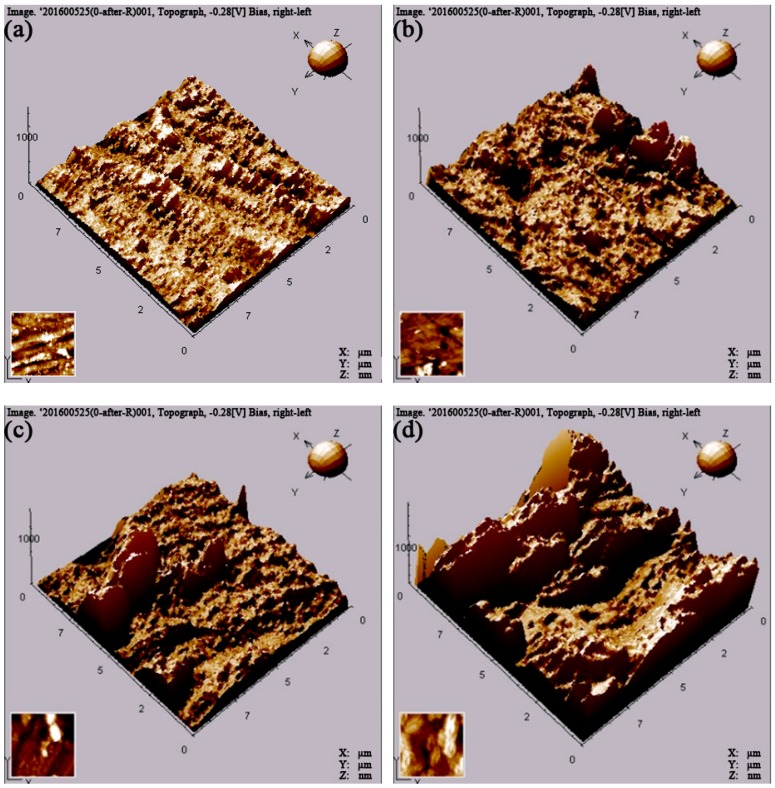
AFM three-dimension images of X80 pipeline steel coupons after removing the corrosion products after 30 days. (**a**) Control; (**b**) SRB; (**c**) SRB + 5 mA/cm^2^ AC; (**d**) SRB + 10 mA/cm^2^ AC.

**Figure 7 materials-12-00144-f007:**
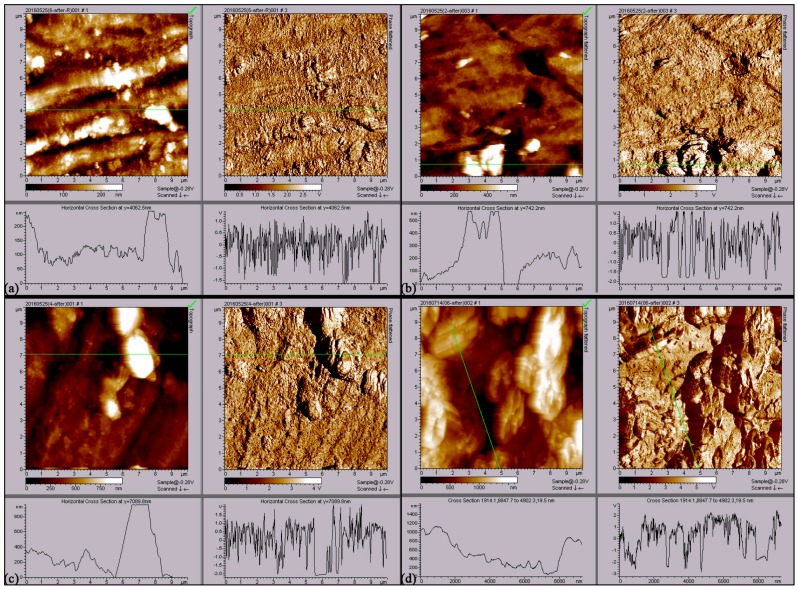
AFM image of X80 pipeline steel coupons after removing the corrosion products after 30 days. (**a**) Control; (**b**) SRB; (**c**) SRB + 5 mA/cm^2^ AC; (**d**) SRB + 10 mA/cm^2^ AC.

**Figure 8 materials-12-00144-f008:**
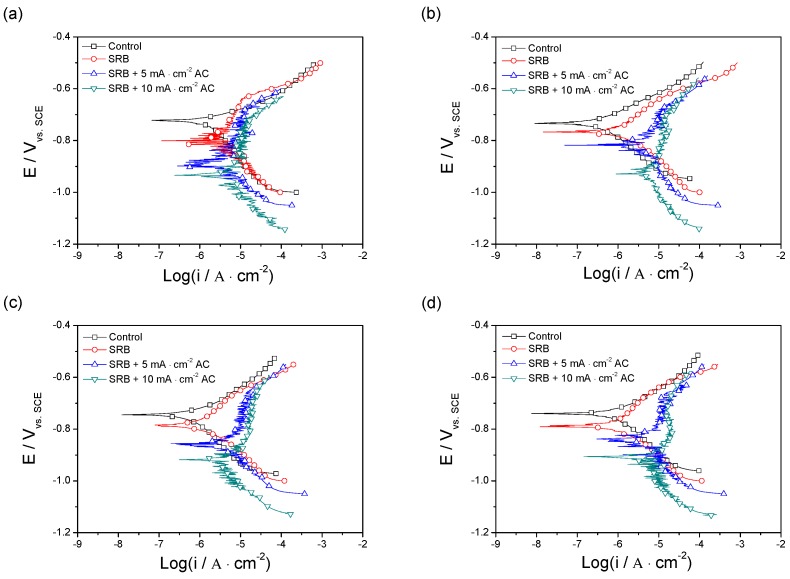
Tafel curves of X80 pipeline steel coupons in the four conditions (Control, SRB, SRB + 5 mA/cm^2^ AC and SRB + 10 mA/cm^2^ AC) after 2 days (**a**), 10 days (**b**), 20 days (**c**), and 30 days (**d**).

**Figure 9 materials-12-00144-f009:**
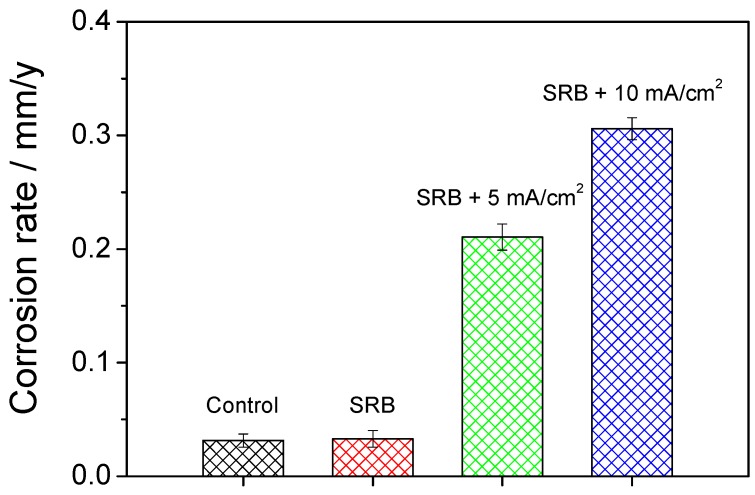
Corrosion rate calculated by weight loss in the four conditions after 30 days.

**Figure 10 materials-12-00144-f010:**
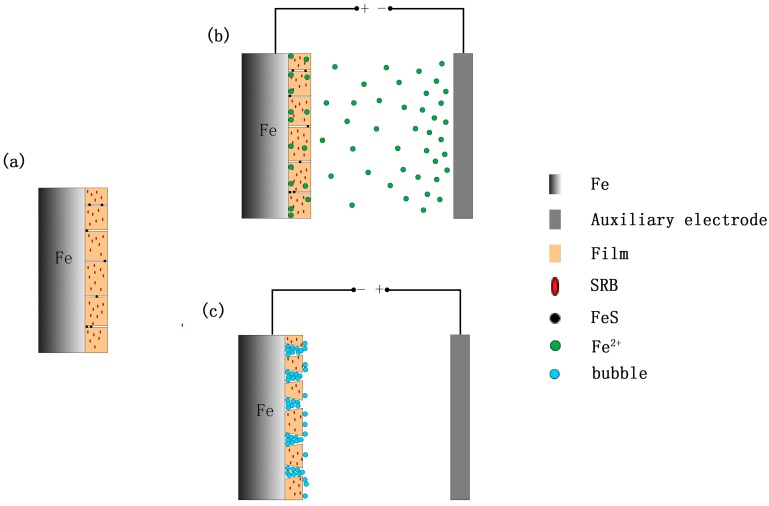
Schematic diagram to explain the effect of AC on corrosion of the X80 pipeline steel (**a**) without AC; (**b**) in the positive half cycles of AC; (**c**) in the negative half cycles of AC).

**Table 1 materials-12-00144-t001:** Chemical compositions of the soil-extract solution (mg/L).

pH	${NO}_{3}^{-}$	Cl^−^	${SO}_{4}^{2-}$	${HCO}_{3}^{-}$	Ca^2+^	Mg^2+^	K^+^	Na^+^	Organics	Total Nitrogen	TOTAL SALT
7.75	46	31	48	234	57	32	2	14	22600	150	464

**Table 2 materials-12-00144-t002:** EDAX results of the corrosion products of X80 pipeline steel coupons in the four conditions (At%).

Element	Control (Position 1)	SRB (Position 2)	SRB + 5 mA/cm^2^ AC (Position 3)	SRB + 10 mA/cm^2^ AC (Position 4)
C	39.32	42.82	46.07	42.98
O	36.02	26.72	23.65	24.73
P	-	7.7	7.23	6.89
S	-	5.8	5.58	3.39
Ca	2.04	1.52	1.48	1.47
Si	0.78	0.64	0.75	0.69
Fe	21.84	14.8	15.24	19.85

**Table 3 materials-12-00144-t003:** AFM analyzing data of X80 pipeline steel coupons after removing the corrosion products and biofilms in the four conditions.

Experimental Condition	Mean Roughness nm	RMS Roughness nm	R_P-V_ nm
Control	43 ± 3	58 ± 4	582 ± 21
SRB	78 ± 5	114 ± 5	1177 ± 32
SRB + 5 mA/cm^2^ AC	136 ± 9	204 ± 8	1676 ± 46
SRB + 10 mA/cm^2^ AC	278 ± 11	334 ± 13	1984 ± 69

**Table 4 materials-12-00144-t004:** Fitting results of Tafel curves of X80 pipeline steel coupons in the four conditions after different times (days).

Time Days	Experimental Condition	E_corr_ V	I_corr_ μA/cm^2^	β_a_ mV/Decade	β_c_ mV/Decade
2	Control	−0.7219	1.67	65	196
	SRB	−0.8069	2.32	190	120
	SRB + 5 mA/cm^2^ AC	−0.8771	2.74	265	123
	SRB + 10 mA/cm^2^ AC	−0.9442	3.99	357	148
10	Control	−0.7328	0.30	83	118
	SRB	−0.7656	1.09	133	126
	SRB + 5 mA/cm^2^ AC	−0.8148	4.12	253	230
	SRB + 10 mA/cm^2^ AC	−0.9347	5.86	485	268
20	Control	−0.7431	0.73	99	169
	SRB	−0.7834	1.11	166	120
	SRB + 5 mA/cm^2^ AC	−0.8613	4.63	312	165
	SRB + 10 mA/cm^2^ AC	−0.9427	5.94	449	172
30	Control	−0.7379	0.90	91	143
	SRB	−0.7854	0.94	163	121
	SRB + 5 mA/cm^2^ AC	−0.8498	6.02	388	212
	SRB + 10 mA/cm^2^ AC	−0.9202	8.75	459	347

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
