# Peer review of "Effect of Alternating Current and Sulfate-Reducing Bacteria on Corrosion of X80 Pipeline Steel in Soil-Extract Solution"

_materials, 2019, doi:10.3390/ma12010144_

Round 1

Reviewer 1 Report

The abbreviation SRB's meaning should be put in the title and not only in the Keywords. 

Please use the same words for the same physical quantity, like Fardaic OR Faradic (?)

Is it correct to nam the technique EDXA OR EDAX ?

In line 121 the unit could be modified (Too big dot.)

In line 523 below the Figure 5 the ending of the sentence is missing.

A last check of the language would improve the text. Is it correct e.g.:  It can been seen... OR    It can be seen...   ?

Author Response

The abbreviation SRB's meaning should be put in the title and not only in the Keywords. 

We revised it.

Please use the same words for the same physical quantity, like Fardaic OR Faradic (?)

We revised it.

Is it correct to nam the technique EDXA OR EDAX ?

We revised it.

In line 121 the unit could be modified (Too big dot.)

We revised it.

In line 523 below the Figure 5 the ending of the sentence is missing.

We revised it.

A last check of the language would improve the text. Is it correct e.g.:  It can been seen... OR    It can be seen...   ?

We revised it.

Reviewer 2 Report

There are a few issues that need to be addressed:

The SEM needs to be described in the materials and method.

The EDXA results are unclear. First of all this I assume this is SEM-EDS (or SEM-EDX). Then what surface did you analyse? I imagine all of the given results could be taken from one surface, if different points were analysed. Secondly EDS is not the best method for analysing light elements like C, especially a "dirty" surface is analysed. So i find this part along with table 2 unconvincing.

If you took samples for planktonic cell counting, did you also make chemical analysis for the increase of Fe in the solution?

The figures are a pressing issue.

Figure 2, the choice of scale is bad. For example the y axle goes from 120x10e6 to negative -15xe6. The negative numbers are nonsense, and the changes you are trying to show are logarithmic, a different scale or a simple table could be better. Also a whole graph for showing three numbers in the sessile SRB (only for 30 days) is annoying and a waste of a figure.

Figure 3: The small corner figures are simply too small and their scalebars are unreadable. Please improve the situation.

Figure 7: What is the point of making almost identical images with almost identical profiles?

Figure 8: Improve  the  Tafel curve diagrams, the points make them harder to read,

Figure 10: FeS point are so small I had to look carefully twice to se where they were supposed to be, please improve.

The state of the steel is important, is it normalised, soft annealed, cold deformed?

Overall the excessive use of abbreviations is annoying, the reader needs to constantly check for the meaning in previous pages. 

Also you mention that SRB reduce the corrosion by protecting the steel with FeS, but I feel this isn't clearly deemonstrated by the results. 

Author Response

There are a few issues that need to be addressed:

The SEM needs to be described in the materials and method.

We have described the SEM method in “2.6 Morphological characterization”.

The EDXA results are unclear. First of all this I assume this is SEM-EDS (or SEM-EDX). Then what surface did you analyse? I imagine all of the given results could be taken from one surface, if different points were analysed. Secondly EDS is not the best method for analysing light elements like C, especially a "dirty" surface is analysed. So i find this part along with table 2 unconvincing.

The reviewer is right, EDAX is not good method for analyzing element C because of contamination. In this paper, we just analyzed element S and P in order to characterizing SRB metabolism.

If you took samples for planktonic cell counting, did you also make chemical analysis for the increase of Fe in the solution?

The reviewer is right, ferrous concentration should be measured because ferrous ions have an important effect on corrosion of the steel, however, we didn’t do it because of the following reasons: (1) It is our faults. We took only 5 ml solution for SRB cell counts, so we had no solutions to measure the other ions. (2) We don’t find a better method that can accurately measure the ion concentrations because of interference of sulfide, e.g., FeS. In future experiments, we will try to measure other ions (Fe2+, Fe3+, S2-) by finding better measuring methods.

The figures are a pressing issue.

Figure 2, the choice of scale is bad. For example the y axle goes from 120x10e6 to negative -15xe6. The negative numbers are nonsense, and the changes you are trying to show are logarithmic, a different scale or a simple table could be better. Also a whole graph for showing three numbers in the sessile SRB (only for 30 days) is annoying and a waste of a figure.

We modified it, and Fig.2 was divided into two figures.

Figure 3: The small corner figures are simply too small and their scalebars are unreadable. Please improve the situation.

We modified them.

Figure 7: What is the point of making almost identical images with almost identical profiles?

We modified them.

Figure 8: Improve  the  Tafel curve diagrams, the points make them harder to read,

We modified them.

Figure 10: FeS point are so small I had to look carefully twice to se where they were supposed to be, please improve.

We modified them.

The state of the steel is important, is it normalised, soft annealed, cold deformed?

We modified them.

Overall the excessive use of abbreviations is annoying, the reader needs to constantly check for the meaning in previous pages. 

The reviewer is right, there are seven abbreviation, AC, SRB, SES, AV, CP, MIC, DO, and it is too much. We deleted four, AV, CP, MIC, DO. We kept other three abbreviations, AC, SRB and SES, in order to writing fluently.

Also you mention that SRB reduce the corrosion by protecting the steel with FeS, but I feel this isn't clearly deemonstrated by the results. 

“It can also been seen from table 4 that, for the condition without the AC current, the corrosion current density of steel coupon decreases with time. It is due to the reason that a layer of composite film containing the corrosion products, biofilm and metabolic sulfide is formed on the surface of steel coupon. This film is dense, compact and protective, as shown in Fig. 3. This result is consistent with Sheng and González’s findings [29,30]. It is well known that the film containing FeS sometimes offers protection against the corrosion [31,32]. Castaneda and El Mendili et al. found iron sulfide could enter into pores of corrosion product film, and with gradual accumulation of the iron sulfide the film got denser and denser, which decreased the corrosion rate [33,34]. ”. What we mean is that the corrosion product layers with FeS is protective in the condition with only SRB. From Table 4, it can be seen that Icorr is 2.32μA/cm2 on the second day, decreasing to 1.09μA/cm2 on the tenth day, so we concluded that the composite films with FeS could protect the steels.